# Untargeted Lipidomics of Vesicular Stomatitis Virus-Infected Cells and Viral Particles

**DOI:** 10.3390/v14010003

**Published:** 2021-12-21

**Authors:** Katherine E. Havranek, Judith Mary Reyes Ballista, Kelly Marie Hines, Melinda Ann Brindley

**Affiliations:** 1Department of Infectious Diseases, College of Veterinary Medicine, University of Georgia, Athens, GA 30602, USA; kthavranek@gmail.com (K.E.H.); jreyesb@uga.edu (J.M.R.B.); 2Department of Chemistry, Franklin College of Arts and Sciences, University of Georgia, Athens, GA 30602, USA; 3Department of Infectious Diseases, Department of Population Health, College of Veterinary Medicine, University of Georgia, Athens, GA 30602, USA

**Keywords:** vesicular stomatitis virus, VSV, untargeted lipidomics

## Abstract

The viral lifecycle is critically dependent upon host lipids. Enveloped viral entry requires fusion between viral and cellular membranes. Once an infection has occurred, viruses may rely on host lipids for replication and egress. Upon exit, enveloped viruses derive their lipid bilayer from host membranes during the budding process. Furthermore, host lipid metabolism and signaling are often hijacked to facilitate viral replication. We employed an untargeted HILIC-IM-MS lipidomics approach and identified host lipid species that were significantly altered during vesicular stomatitis virus (VSV) infection. Many glycerophospholipid and sphingolipid species were modified, and ontological enrichment analysis suggested that the alterations to the lipid profile change host membrane properties. Lysophosphatidylcholine (LPC), which can contribute to membrane curvature and serve as a signaling molecule, was depleted during infection, while several ceramide sphingolipids were augmented during infection. Ceramide and sphingomyelin lipids were also enriched in viral particles, indicating that sphingolipid metabolism is important during VSV infection.

## 1. Introduction

Vesicular stomatitis virus (VSV) is a non-segmented negative-stranded RNA virus belonging to the *Rhabdoviridae* family. VSV infects a range of mammals, including horses, livestock, and humans, although disease in humans is mild. VSV encodes five proteins: nucleocapsid (N), phosphoprotein (P), matrix (M) protein, glycoprotein (G), and large (L) viral polymerase. Recombinant VSV vectors are commonly used in molecular biology laboratories, vaccine development, gene therapy, and oncolytic virotherapy [1,2]. While many of the host proteins involved in virus production have been identified, the lipid components utilized by the virus, as well as the viral impact on the host lipidome, remain to be determined. Investigating the contribution of host lipids during viral infection is critically important to understanding pathogenesis and will aid in further development and production of VSV-based vaccines and therapeutics.

Viruses utilize host lipids at several stages of their lifecycle. Enveloped viruses undergo membrane fusion at host plasma or endosomal membranes in order to gain entry into the host cell. The VSV single-transmembrane glycoprotein G interacts with cellular receptors from the low-density lipoprotein receptor (LDL-R) family and mediates low-pH-induced membrane fusion following clathrin-mediated endocytosis [3,4,5,6]. There is evidence to suggest that G interacts with anionic lipids in the late endosome, which facilitates efficient membrane fusion [7,8]. Host lipid metabolism can also serve as an important source of energy required to fuel viral replication [9,10,11]. During VSV infection, host mitochondrial short-chain enoyl-CoA hydratase (ECHS), which catalyzes the β-oxidation pathway of fatty acids, was shown to be important for viral replication [12].

The integrity of the host lipid bilayer is of vital importance for efficient virus production. Enveloped viruses derive their lipid exterior during the budding process when the virus enrobes itself in host-derived membranes. All viral lipids are hijacked from host membranes, but the lipid constituents and budding locations of different viruses vary. VSV forms bullet-shaped particles that bud from the plasma membrane as the M protein interacts with both the inner leaflet of the membrane and the ribonucleoprotein complex [13]. M contains two N-terminal late budding domains, PPPY and PSAP, which recruit ESCRT1 components that mediate fission of the viral and host membranes [14,15,16]. There is conflicting evidence concerning the importance of membrane microdomains during VSV assembly and budding. VSV is commonly used to generate pseudotyped viruses since the progeny VSV core will readily incorporate the envelope glycoproteins of other viruses. It has been suggested that lipid rafts mediate the phenotypic mixing that is required for generating pseudotyped particles [17]. However, VSV G lacks traits commonly correlated with raft-associated proteins, and G has been shown to only weakly associate with raft domains [18,19]. While VSV particles have been shown to incorporate raft-associated proteins and lipids, the most recent lipidomic evidence suggests that the VSV envelope mirrors that of the producer cells [20,21,22].

In this study, we performed untargeted HILIC-IM-MS lipidomics during VSV infection in mammalian cells to investigate changes in host lipids and lipid incorporation into viral particles. HILIC-IM-MS offers the separation of lipid species from many different classes, including glycerophospholipids and sphingolipids, based on their headgroup polarities [23]. Over the course of VSV infection, the host lipidome was altered in a manner that indicated a shift towards lipid synthesis rather than metabolism. Ontological enrichment analysis suggested that these lipid changes alter host membrane properties. Consistent with an alteration in membrane dynamics was the discovery that the conical-shaped lysophosphatidylcholine lipids were among the most significantly decreased lipid species late in infection, while ceramides were increased. Lipid analysis of VSV particles indicated that the fatty acyl composition of particle lipids was similar to that of cellular lipids, but that several sphingolipid species were enriched in viral particles.

## 2. Materials and Methods

### 2.1. Materials

HPLC grade solvents (water, acetonitrile, methanol, and chloroform) and ammonium acetate were purchased from Thermo Fisher Scientific (Waltham, MA). A mixture of phospholipids, glycerolipids, and sphingolipids representative of mammalian lipid composition was prepared as described previously [23] and used as a HILIC retention time reference.

### 2.2. Cell Culture and Viruses

Vero (Vervet monkey kidney) cells stably expressing SLAM were maintained at 37 °C and 5% CO_2_ in Dulbecco’s modified Eagle’s medium (DMEM) supplemented with 5% FBS. Recombinant VSV encoding the GFP or nano luciferase fused to a destabilization domain PEST (nLuciP) reporter gene was recovered from transfecting cells with the four plasmid VSV rescue system (a gift from Michael Whitt, University of Tennessee, Kerafast, 7753828) as previously described [24].

### 2.3. Viral Infection for Lipidomic Analysis

Vero/hSLAM cells were grown to 90% confluence in 10 cm plates and mock-infected or infected with VSV at an MOI of 0.1 in serum-free DMEM for 1 hr at 37 °C 5%CO_2_. After infection, the media were replaced with DMEM supplemented with 5% FBS and cells were maintained for 18 h. Supernatants were collected, and cells were rinsed in PBS, pelleted (800× *g*, 3 min), and stored at −80 °C until lipid extraction. Two biological replicates were performed, and each replicate included four infected and four uninfected 10 cm plates to serve as technical replicates for lipid extraction, total *n* = 16. Virus or extracellular material found in the supernatants from technical replicates was pooled and concentrated by ultracentrifugation. Samples were pelleted through a 20% sucrose cushion (100,000× *g*, 2 hr, 4 °C), resuspended in 0.5 mL PBS, and frozen at −80 °C until use.

### 2.4. Sample Preparation for Lipidomic Analysis

Cell pellets were resuspended in 0.5 mL of water and transferred to glass round-bottom tubes. Lipids were extracted according to a modified version of the Bligh and Dyer method [25]. Briefly, 2 mL of pre-chilled 1:2 chloroform/methanol was added to the pellet resuspension and vortexed for 5 min. Then, 0.5 mL of chilled water and 0.5 mL of chilled chloroform were added, and samples were vortexed 60 sec followed by centrifugation (1000× *g*, 10 min, 4 °C). The organic layer was transferred into new glass tubes and dried in a SpeedVac for 1 hr. Dried extracts were frozen at −80 °C until use. For HILIC-IM-MS analysis, dried extracts were reconstituted in 0.5 mL 1:1 chloroform/methanol. Ten microliters of cellular material or 100 µL of pelleted supernatant material was dried in a SpeedVac and resuspended in 100 µL 2:1 acetonitrile/methanol.

### 2.5. HILIC-IM-MS and Lipidomic Data Analysis

Extracts were characterized by hydrophilic interaction liquid chromatography and traveling wave ion mobility-mass spectrometry in both positive and negative ionization modes. A CORTECS HILIC column (2.1 × 100 mm, 1.6 µm, Waters) was used to separate lipids by head group polarity over a 12 min gradient [23]. The column effluent was connected to the electrospray ionization source of a Waters Synapt XS, as described previously [23]. Peak picking, normalization, and alignment were performed with Progenesis QI (Nonlinear Dynamics). Data from the supernatant samples were used to determine the lipid composition of the budded VSV particles. Data from the cell lysates were processed with Progenesis QI (Nonlinear Dynamics) and MetaboAnalyst [26] to identify statistically significant features between the mock- and VSV-infected cells. Lipid identifications were made against the LipidMaps CompDB tool with a mass accuracy threshold of 10 ppm. Ontological analysis was performed with LION/web [27]. Full data tables are available in the Appendix A; cellular lipids (Appendix A) and particle lipids (Appendix A).

### 2.6. VSV-G Kinetics with LysoPC Supplementation

Cell-to-cell spread of VSV-G in Vero cells was determined using Endurazine substrate (Promega, Madison, WI, USA) following manufacturer’s specifications. Vero cells were plated in a black-walled, clear-bottom 96-well plate at a density of 3 × 10^5^ cells/mL with DMEM 5% FBS. Once attached to the plate, cells were starved from lipids by replacing the media with serum-free DMEM. After 20 h, media were substituted with DMEM 5% Lipoprotein Depleted FBS (Cat #880100-1, Kalen Biomedical LLC, Germantown, MD, USA). Cells were infected with 50 µL of VSV-G-nLuciPest at an MOI of 0.04 using serum-free DMEM. One hour after infection, media were removed and replaced with 200µL of phenol red-free DMEM supplemented with 5% Lipoprotein Depleted FBS, 25 mM HEPES, 1:200 Endurazine and treated with LysoPC 16:0 (SKU #855675P, Avanti Polar Lipids, Birmingham, AL, USA) or 18:1 (SKU #845875P, Avanti Polar Lipids, Birmingham, AL, USA) 0.5 µM or 5 µM. The plate was moved into a pre-warmed (37 °C) GloMax Explorer (Promega, Madison, WI, USA), and luminescence was measured every 10 min for a period of 24 h.

### 2.7. VSV-G Titers with LysoPC Supplementation

To determine viral titers, VeroS cells were plated in a 24-well plate and treated in a similar manner to VSV-G kinetic assay. Twenty-four hours post-infection, supernatants were collected, and amount of infectious viral particles was determined by infecting VeroS cells with serial dilutions of the virus to calculate the median tissue culture infectious dose (TCID50) using the Spearman–Karber method.

## 3. Results

### 3.1. HILIC-IM-MS Lipidomic Profiling Reveals Major Lipid Classes Change in Response to VSV Infection

To better understand host lipid dynamics during VSV infection, untargeted HILIC-IM-MS lipidomics was performed on lipids extracted from Vero/hSLAM cells infected or mock-infected with VSV at an MOI of 0.1 for 18 hr (Figure 1A). Supernatants were also collected for lipidomic analysis of particles derived from VSV infected or mock-infected cells (Figure 1A). From the positive mode dataset, 590 lipid features with ANOVA *p* ≤ 0.05 were identified (Figure 1B). The major classes of membrane-forming lipids in mammalian cells ionize more efficiently in the positive electrospray ionization mode; therefore, data-analysis efforts focused on the positive mode data set to determine lipids altered in VSV infection.

Of the significantly altered features, a total of 101 lipid species were identified using the LipidMaps CompDB tool. Putative lipid IDs were evaluated alongside the HILIC retention times and the data-independent MS/MS spectra in the negative mode to confirm lipid class and fatty acyl composition, respectively (Figure 2A). Ion mobility extracted ion chromatograms of mock- and VSV-infected cells depict peaks corresponding to the major lipid classes that were identified (Figure 2A). Integration of chromatographic peak areas shows that there are similar proportions of lipid classes in both mock- and VSV-infected cell lysates (Figure 2B).

A more detailed analysis of the individual lipid species within the major lipid classes analyzed in the positive ionization mode data set revealed trends during infection. Phosphatidic acids (PAs), phosphatidylcholine lipids (PCs), and phosphatidylinositol (PI) showed a general trend towards increased abundance during infection (Figure 3A). PAs that were elevated in VSV-infected cells were increased along with PCs and PIs with the same fatty acyl compositions (Figure 3A, Appendix A). However, other glycerophospholipid species, including lysophosphatidylcholines (LysoPC/LPCs) and phosphatidylglycerols (PGs), displayed an overall trend towards decreased abundance in infected cells (Figure 3B). The collective elevation in PC, PI, and PA lipids with the combined decrease in LPC indicates that there may be more lipid synthesis than metabolism occurring in infected cells at this time post-infection. Phosphatidylethanolamine (PE) and lysophosphatidylethanolamine (LysoPE/LPE) species did not display a pattern towards increased or decreased abundance (Figure 3C). The ceramide (Cer), hexosylceramides (HexCers), and sphingomyelin (SM) sphingolipids with no double bonds were elevated in VSV-infected cells, while other species were decreased (Figure 3D). Enrichment analysis using LION suggests that these alterations collectively result in a change to the intrinsic membrane curvature and bilayer thickness during VSV infection (Figure 4A). Fourteen individual lipid species were altered above a threshold of fold change >1.5 with *p* < 0.05 (Figure 4B). Interestingly, the overall trend was towards an increase in ceramide sphingolipids and a decrease in lysophosphatidylcholine as well as the long-chain polyunsaturated fatty acid eicosadienoic acid (EDA).

### 3.2. VSV Cell-to-Cell Spread Is Enhanced by LPC (18:1)

To further explore the function of LPC during VSV infection, we examined the cell-to-cell spread kinetics of VSV expressing nLuciPest by monitoring the production of luminescence and quantified the amount of infectious viral particles at 24 h. For this, we serum-starved Vero cells, infected using delipidated FBS and treated with increasing amounts of lysoPC 16:0 or 18:1. Supplementation of LPC 16:0 showed no significant differences in cell-to-cell spread (Figure 5A) nor the amount of infectious viral particles (Figure 5B). However, supplementation with LPC 18:1 exhibited a modest twofold increase in cell-to-cell spread (Figure 5C). No significant differences were seen in titers between treatments at 24 h, possibly due to high titers overall among groups (Figure 5D).

### 3.3. VSV Virions Are Enriched in Sphingolipids

In order to investigate the lipid composition of VSV particles derived from Vero/hSLAM cells, supernatants were collected from infected and mock-infected cells and concentrated by ultracentrifugation. The lipid composition was determined by HILIC-IM-MS. As expected, HILIC profiles show large differences between the VSV and mock-infected supernatants (Figure 6A,B). However, it is important to note that exosomes and other lipid-derived vesicles expelled from mock- and VSV-infected cells were not excluded from this analysis. The composition of lipids derived from the VSV-infected cell supernatants was largely consistent with the composition of lipids in the cell lysates (Figure 6C). However, the VSV particles were enriched in the Cer, HexCer, and SM sphingolipids relative to the VSV-infected cell lysates (Figure 6C, Appendix A). This enrichment is evident from the comparison of PC abundance, the major lipid in the cell lysates, relative to the abundances of the sphingolipids in the VSV particles.

## 4. Discussion

Here, we apply a HILIC-IM-MS-based untargeted lipidomics approach to investigate host and viral lipids during VSV infection. Analysis of the host lipidome revealed alterations in several major lipid classes during infection. Increased levels of PC, PI, and PA glycerophospholipids with a concurrent decrease in the most abundant cellular species of lysolipid, LPC, indicates increased lipid synthesis at the expense of lipid metabolism during VSV infection.

It is becoming increasingly clear that many viruses hijack and modify host lipid synthesis and cellular metabolism to enhance their replication. Several DNA viruses, including Epstein–Barr virus (EBV), Kaposi’s sarcoma-associated herpesvirus (KSHV), human cytomegalovirus (HCMV), and Vaccinia virus (VACV), have been shown to alter host lipid metabolism [28,29,30,31,32,33]. Replication of positive-sense RNA viruses Dengue virus (DENV), West Nile virus (WNV) and Japanese encephalitis virus (JEV) from the Flaviviridae family, and severe acute respiratory syndrome coronavirus 2 (SARS-CoV-2) [34], is critically dependent upon host lipid metabolism [11,35,36,37]. In contrast, the importance of lipids in negative-sense RNA virus replication is not well characterized. However, fatty acid synthesis has been shown to be critically important for IAV infection [38,39]. Fatty acid synthesis is also important for VSV infection, since the knockdown of mitochondrial short-chain enoyl-CoA hydratase (ECHS), which catalyzes the β-oxidation pathway of fatty acids, decreases VSV replication [12]. This work indicates that a prototypical, single-stranded, negative-sense RNA virus has the capacity to alter host lipid production, and warrants further investigation into the mechanism(s) by which lipid synthesis is modified during infection.

LPCs were identified as the most significantly decreased host lipid species during VSV infection. This contrasts with lipidomic data from positive-sense RNA viruses such as human coronavirus 229E (HCoV-229E) and WNV that exhibited elevated LPC during infection [40,41]. During WNV infection, LPC production contributes to the formation of curved membranous replication complexes characteristic of positive-sense RNA viruses [40]. LPC contributes to membrane curvature due to its inverted-cone shape, which arises when one of the fatty acid chains is removed from phosphatidylcholine. The enzymatic activity of phospholipase A2 (PLA_2_) catalyzes the hydrolysis of the *sn*-2 fatty acid ester to generate LPC. Conversely, the enzymatic activity of lysophosphatidylcholine acetyltransferase (LPCAT) acts to reacylate LPC to form PC. The use of phospholipase enzymes to modify the shape of membrane lipids has also been shown to directly influence the efficiency of membrane fusion, which indicates that altered LPC levels could impact viral entry and/or egress [42,43,44,45]. The addition of LPC (18:1) modestly enhanced cell-to-cell spread of VSV in a multi-cycle infection, suggesting that depletion during infection may decrease the efficiency of spread. However, additional LPC (16:0) did not significantly alter spread even though both LPC (18:1) and LPC (16:0) were depleted from infected cells. In addition to its importance in membrane composition and curvature, LPC also acts as a signaling molecule capable of regulating apoptosis, inflammation, and oxidative stress [46]. Future studies will further characterize the changes in replication with additional LPC or reduced levels of LPC to determine if VSV requires specific LPCs directly or for cell-signaling events.

The ceramide class of sphingolipid was the most significantly increased host lipid species during VSV infection. Ceramides are critical intermediates of sphingolipid metabolism, consisting of a fatty acyl chain bound to a sphingosine base. Due to their hydrophobic nature and low polarity, ceramides are insoluble in water and therefore are not found freely in biological fluids [47]. Within membranes, ceramides serve to enhance both lateral membrane organization and the order of phospholipid acyl chains [48,49]. Ceramide synthesis can take place via several different pathways: de novo synthesis in the ER, sphingomyelinase action at the plasma membrane, and hydrolysis of complex sphingolipids [50]. Ceramides also act as bioactive lipids capable of regulating cell proliferation, apoptosis, and disease [51].

Over the course of a viral infection, ceramides engage in critical but diverse roles. During IAV infection, ceramide accumulates in human lung epithelial cells via the de novo synthesis pathway and acts in an antiviral capacity [52]. In contrast, HCV infection enhances ceramide and sphingomyelin production to enhance viral replication [53]. Other positive-sense RNA viruses appear to utilize ceramide differently during infection. West Nile Virus Kunjin strain redistributes ceramide to its replication complex, and depletion of ceramide negatively impacted virus production, but Dengue replication was actually enhanced by ceramide depletion [54]. Interestingly, along with the increase in ceramide lipids in infected cells, lipidomic profiling of viral particles showed that sphingomyelins and ceramides made up a significant proportion of the identified lipid species in VSV particles. Further work should address the relevance of increased ceramide production, sphingolipid metabolism, and their role in productive VSV infection.

Lipidomic analysis of supernatants from VSV infected cells indicated an increased proportion of sphingomyelins and ceramides relative to infected cell lysates. Although this implies that VSV may bud through sphingolipid microdomains called lipid rafts, the HILIC method does not capture cholesterol, which is a key component of membrane rafts. Furthermore, many of the sphingolipids and glycerophospholipids within viral particles exhibited degrees of unsaturation, yet primarily saturated sphingolipids and PC lipids associate with cholesterol to partition into lipid rafts [55,56]. Existing studies present conflicting evidence regarding sphingolipid enrichment in VSV particles [22,57]. One study showed that when VSV budded through the basolateral plasma membrane in polarized epithelial MDCK cells, sphingomyelins were enriched in the viral particles [57]. When the lipid components of VSV particles budding from BHK cells were analyzed using shotgun lipidomics, the authors noted a depletion of ceramide in viral particles (22). The objective of the study was to compare the viral lipid composition with that of the purified host cell plasma membrane in order to determine whether viral lipids were selectively incorporated into particles. To achieve the objective, they chose to infect cells with the lowest dose of virus possible to infect 80% of cells in the first round of replication. Notably, our study differs from the latter in that we used an MOI of 0.1 over the course of an 18 h infection, with the goal of ensuring that every cell is infected while also allowing sufficient time to capture virus-driven changes in the host lipid metabolism. We also did not enrich the plasma membrane but examined the total cellular lipidome to capture global cellular changes induced by the virus. The variation in cell type and the number of infected cells, as well as the stage of infection, could account for the differing viral particle composition. Overall, the changes in host lipidome and viral particle composition strongly suggest that future studies should aim to unravel the importance of sphingolipid metabolism during VSV infection.

## Figures and Tables

**Figure 1 viruses-14-00003-f001:**
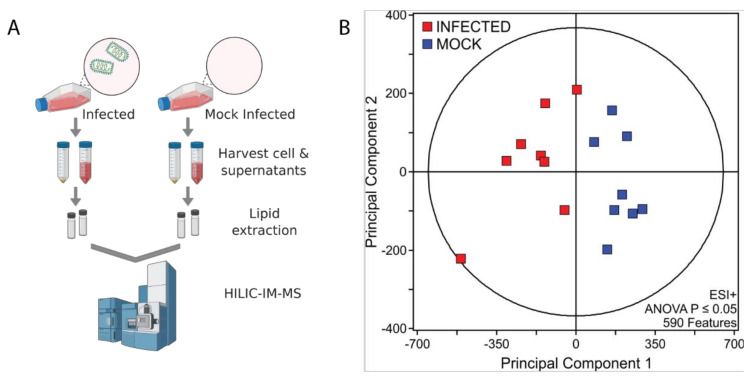
HILIC-IM-MS lipidomic profiling. (**A**) Depiction of workflow. Created in BioRender.com; (**B**) Data sets comparing mock-infected and infected cells were filtered by ANOVA *p*-value ≤ 0.05.

**Figure 2 viruses-14-00003-f002:**
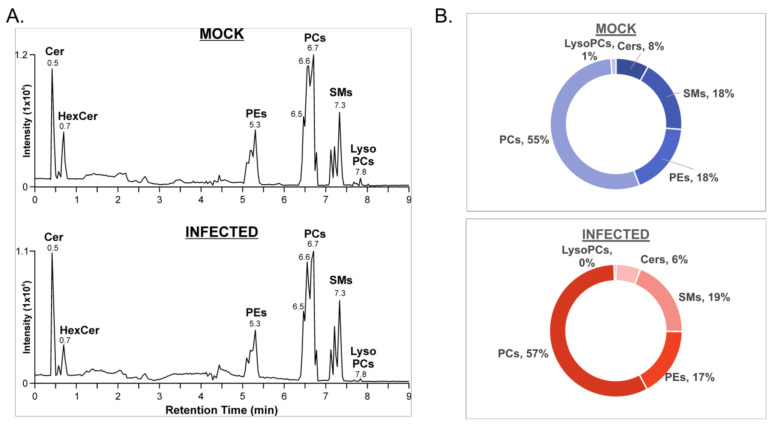
Lipid identification from the positive ionization mode data set for mock- and VSV-infected cells. (**A**) Ion mobility extracted ion chromatograms of mock- and VSV-infected cells from positive ionization mode; (**B**) Proportions of major lipid classes identified in positive ionization mode from mock- and VSV-infected cells determined by manual integration of chromatographic peak areas.

**Figure 3 viruses-14-00003-f003:**
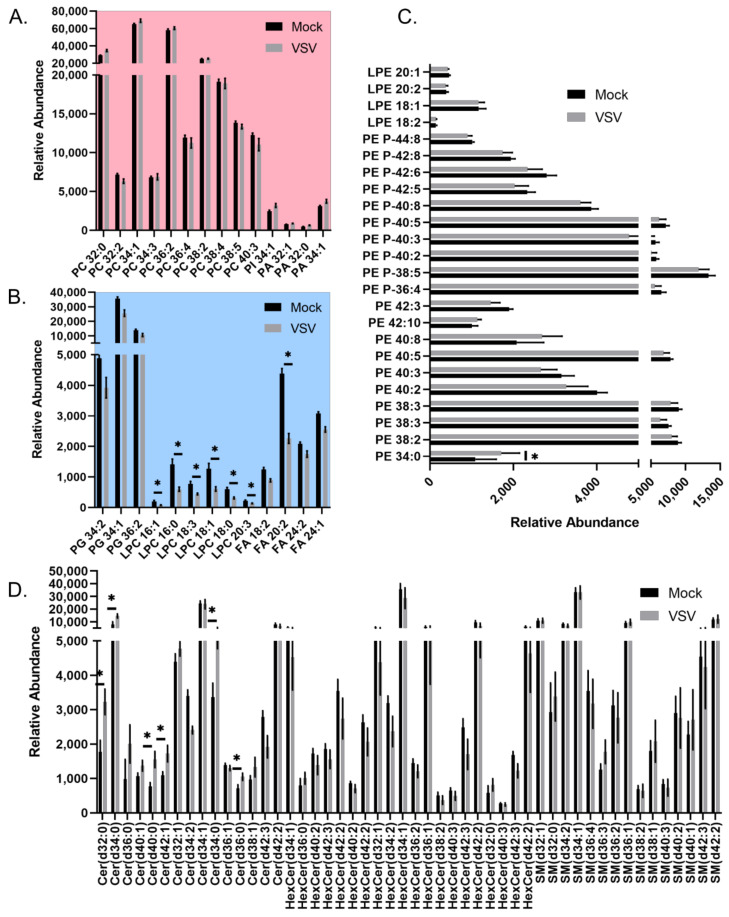
All lipid species identified in the positive mode data set for mock- and VSV-infected cells. Relative abundance of all identified lipids belonging to (**A**) PC, PI, PA classes, which demonstrated overall increased abundance; (**B**) Belonging to PG, LPC, FA classes, which demonstrated overall decreased abundance; (**C**) LPE and PE lipids; (**D**) Cer, HexCer, SM classes. Significantly altered lipids described in Figure 4B are denoted with an asterisk. Note, some lipids have the same sum composition (i.e., total # carbons:total # double bonds), but are not structurally identical to level of specific fatty acyl tails and positions. Lipids that are included multiple times appeared at different retention times or had unique *m*/*z* values due the formation of different adducts.

**Figure 4 viruses-14-00003-f004:**
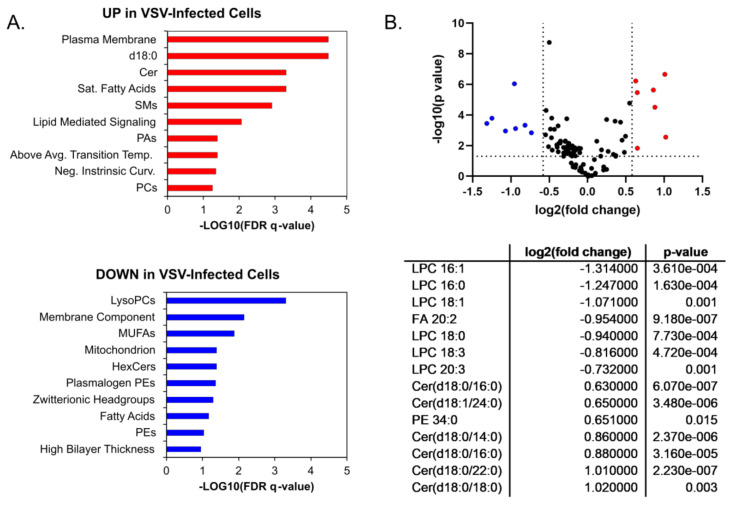
Ontology and significantly altered lipids. Enrichment ontology analysis performed in LION/web for (**A**) Upregulated lipids (in red) and downregulated lipids (in blue); (**B**) Lipid changes displayed according to fold change and p-value threshold significance cutoffs of 1.5 and 0.05, respectively. P-values were determined by Student’s *t*-test (two-tailed, equal variance). Significantly upregulated (red) and downregulated lipid species (blue) are listed in the table in the lower panel.

**Figure 5 viruses-14-00003-f005:**
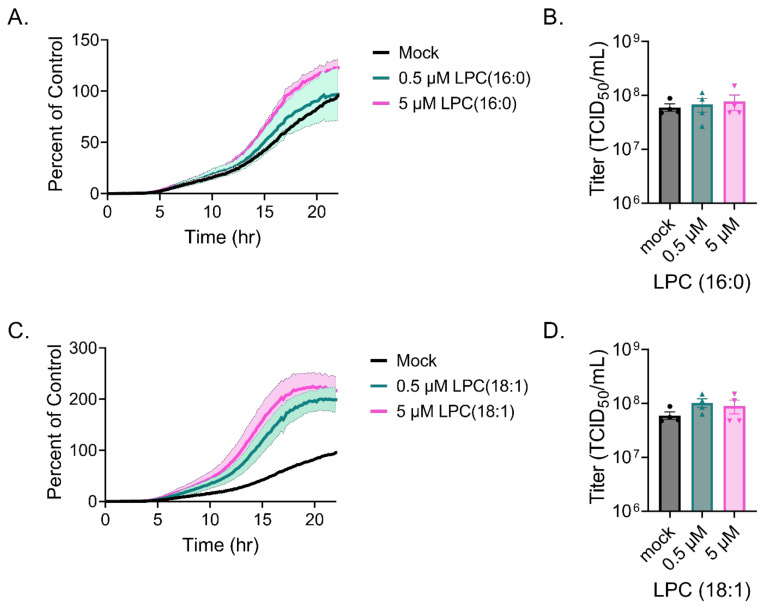
LPC supplementation during VSV infection. Vero cells were infected with VSV-nLuciPest, and cell-to-cell spread kinetics were monitored for 24 h with supplementation of LPC 16:0 (**A**) or 18:1 (**C**). Titers were determined at 24 h after supplementation with LPC 16:0 (**B**) or 18:1 (**D**) by performing serial dilutions and calculating the TCID50 using the Spearman–Karber method.

**Figure 6 viruses-14-00003-f006:**
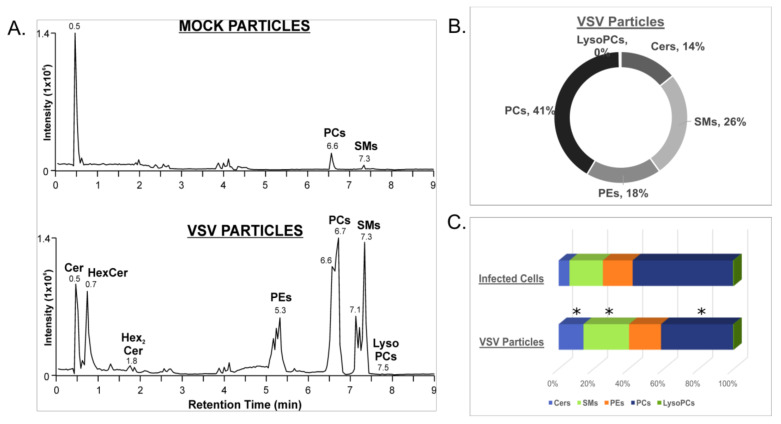
Lipid identification from the positive ionization mode data set for concentrated supernatants from mock- and VSV-infected cells. (**A**) Ion mobility extracted ion chromatograms of mock- and VSV-infected cell supernatants from positive ionization mode; (**B**) Proportions of major lipid classes identified in positive ionization mode from mock- and VSV-infected cell supernatants determined by integration of chromatographic peak areas; (**C**) Comparison of lipid compositions of infected cells and VSV particles derived from the infected cells’ supernatants. Statistically significant changes in lipid class composition are denoted with an asterisk, indicating p-values of less than 0.05 determined by Student’s t-test (two-tailed, unequal variance).

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
