# Peer review of "Untargeted Lipidomics of Vesicular Stomatitis Virus-Infected Cells and Viral Particles"

_viruses, 2021, doi:10.3390/v14010003_

Round 1
Reviewer 1 Report
In this manuscript (ms.) the authors assess changes in total cell lipids at 18 hours post infection of Vero cells cells with VSV as well as the lipid composition of VSV particles released into the culture medium. They report changes in cell lipids at 18 hpi (Figs. 2-4) and enrichment of sphingolipids in virus particles (Fig. 5). The work is generally well-done and well-presented.
Major Comment
- Discussion: The authors should acknowledge that they compared lipid compositions of (relatively crude) VSV particles with total cell lipids, whereas if the reader is not mistaken, Refs. [22] and [57] compared lipid compositions of gradient-banded VSV particles. Also Ref. [22] analyzed purified plasma membrane (vs. total cell membranes as done in the ms. under consideration). These differences are relevant to the discussion of whether VSV buds from rafts.
Minor Comments
- Page 2: This reader is not aware of VSV particles being described as filamentous, just ‘bullet-shaped’. Check on this, but if so amend to simply say ‘bullet-shaped’.
- Page 2: At first usage please spell out what HILIC-IM-MS is an abbreviation for.
- Page 2, bottom, 1st sentence p9, and 6 lines from end of Discussion. Rephrase: as written it sounds as if time points were taken during the course of a VSV infection, but it appears (from Methods) that all samples were taken at 18 hpi.
- Page 5 (under Fig 2): Clarify what is meant by ‘significantly altered features’
- 3: (a) Indicate which changes are statistically significant. (b) Modify Fig. 3 legend, as not all of the comparisons are statistically different (as you make clear on p7 and with Fig. 4).
- Refer to Fig. 4 at the end of sentence (page 7) beginning “Fourteen…” along with cueing to Fig. 3B.
- In text re: Fig. 5 or Fig. 5 legend indicate (if correct; reader inferred from Methods) that these were pooled samples (and was this from 1 or both experiments, 4 or 8 plates). The latter could be clarified in Methods.
- Discussion: “When the lipid components of VSV particles budding from BHK cells were analyzed using shotgun lipidomics, the authors noted a depletion of ceramide in viral particles. The objective of this study was to compare the viral lipid composition with that of the purified host cell plasma membrane.” (a) Provide a reference after ‘viral particles”, presumably [22]. (b) Clarify which study when you write “this study” ([22] or [57]).
Author Response
Thank you for your careful review of the manuscript. We have provided a point-by-point response in the attached document.
Please see the attachment

Reviewer 2 Report
Lipids are essential for many cellular functions and are involved in multiple steps in the viral life cycle. In this paper, Havranek et al analyzed the alteration of the lipid composition of the host cell after VSV infection. They conclude that VSV modifies the lipid profile of the host cell and that sphingolipids are enriched in the viral envelope. Understanding the role of lipids in VSV infection will provide important insights into the study of host-virus interactions. Nevertheless, the results are preliminary and several experiment should be conducted before publication.
Specific comments
- From data presented in Fig. 3, the authors conclude that VSV infection provoques the increase in PC, PI and PA in Vero cells whereas PG, LPC and some fatty acid species decreased. Nevertheless, they do not present statistical analysis to corroborate these conclusions. Moreover, with the axes scales that are used in Figure 3, the small values are not clear. I suggest splitting the axes for the high and low values to better appreciate the small values.
- The authors study the VSV lipid composition (Figure 5) from the infected cell supernatants. As virus may interfere with lipid metabolism, the lipids in the cell supernatant may differ from those of the mock cultures. VSV must be purified from cell cultured before analyzing the viral lipid composition. Moreover, to validate their conclusions, the researchers must include a statistical analysis of data showing differences in lipid composition between cells and VSV particles (Fig. 5C).
- What is the most plausible hypothesis about the function of LPC in VSV infection? The authors do not make it clear in the discussion (page 9). To clarify the issue, I suggest for instance performing further experiments such as LPC supplementation assays and analyzing the effect of phospholipase A2 inhibition on VSV infection.
- The function of SM and ceramides in VSV interaction with the host cell also needs further investigation. I suggest assaying the effect of inhibitors of ceramide biosynthesis on VSV infection and analyzing the VSV binding to lipid rafts by isolation of DRM fraction on discontinuous Optiprep gradients or similar.
- Does the last part of the discussion (page 10) refer to reference 22 (Kalvodova et al., J. Virol. 2009)? It is not clear in the manuscript.
Minor points
- Page 7, lines 14 and 16, read Figure 3A ad Figure 3B and should read Figure 4A ad Figure 4B
- Page 9, line 15. The authors stated, “Fatty acid synthesis is also important for VSV infection”. As the enzyme enoyl-CoA hydratase is involved in b-oxidation, they should say “Fatty acid oxidation or catabolism” instead of synthesis.
Reviewer 3 Report
Dear Editor
I have the following comments on the article: “Untargeted lipidomics of vesicular stomatitis virus infected cells and viral particles”.
This paper determined that during vesicular stomatitis virus (VSV) infection, the host lipidome was altered in a manner that indicated a shift towards lipid synthesis rather than metabolism. The study design and methodology are sound for some extent and the results are well-presented. The following should be considered by authors to improve the quality of the manuscript:
Please improve the language aspects of the manuscript as there are grammatical errors and typos.
Suggestion: results and conclusion should not be placed in the Introduction section (last paragraph).
Although the article is interesting, I think it is important that the authors answer the following questions:
Is it possible that the fetal bovine serum (FBS) with which the infected cells were supplemented could interfere with lipidomic analysis, since it has lipidemic properties?
Due to the extracellular vesicles secreted by virus-infected cells play an important role during infection, How can the authors confirm that the enriched fraction of "viral particles" does not contain extracellular vesicles (exosomes) with viral particle, protein or RNA? Because these vesicles contain a density similar to that of some viral particles.
By signing this letter, I am suggest: Accept after minor revision.
Regards,
Round 2
Reviewer 2 Report
The manuscript has been improved in the second version.
Minor points
Although stated by the authors, these minor points have not been changed in the new version:
- Page 9, lines 205-207, read Figure 3A ad Figure 3B instead of 4A ad Figure 4B
- Second paragraph of the discussion. It is stated, “Fatty acid synthesis is also important for VSV infection”. As the enzyme enoyl-CoA hydratase is involved in b-oxidation, they should say “Fatty acid oxidation or catabolism” instead of synthesis.